# Upregulation of Cell Surface Glycoproteins in Correlation with KSHV LANA in the Kaposi Sarcoma Tumor Microenvironment

**DOI:** 10.3390/cancers15072171

**Published:** 2023-04-06

**Authors:** Sara R. Privatt, Owen Ngalamika, Jianshui Zhang, Qinsheng Li, Charles Wood, John T. West

**Affiliations:** 1School of Biological Sciences, Nebraska Center for Virology, University of Nebraska-Lincoln, Lincoln, NE 68588, USA; 2Department of Interdisciplinary Oncology, Louisiana Cancer Research Center, Louisiana State University Health Sciences Center-New Orleans, New Orleans, LA 70112, USA; 3University Teaching Hospital, University of Zambia School of Medicine, Lusaka 10101, Zambia

**Keywords:** Kaposi sarcoma, Kaposi sarcoma-associated herpesvirus, biomarker, tumor microenvironment

## Abstract

**Simple Summary:**

Despite the widespread use of ART and decreased HIV incidence, Kaposi Sarcoma (KS), predominantly recognized as an AIDS-associated malignancy, continues to be highly prevalent in sub-Saharan Africa, parts of the Mediterranean, and parts of South America. Currently, there are no preventative measures or curative therapies for KS. The treatment is primarily pleiotropic chemotherapy, but the outcomes are inconsistent. Using our previously published transcriptomic profiles from KS lesions, we have identified potential biomarkers, therapeutic targets, and cell lineage determinants that are upregulated in expression and associated with KSHV infection. The protein expression of four surface-associated glycoproteins, FLT4, KDR, UNC5A, and ADAM12, was validated in human KS lesions and KSHV-infected cell line-derived mouse xenografts. The co-localization of CD34 and Prox-1, which are distinct endothelial lineage markers, suggests that cells within KS lesions are of a mesenchymal or progenitor phenotype rather than of a singular endothelial lineage.

**Abstract:**

HIV-associated epidemic Kaposi sarcoma (EpKS) remains one of the most prevalent cancers in sub-Saharan Africa despite the widespread uptake of anti-retroviral therapy and HIV-1 suppression. In an effort to define potential therapeutic targets against KS tumors, we analyzed previously published KS bulk tumor transcriptomics to identify cell surface biomarkers. In addition to upregulated gene expression (>6-fold) in the EpKS tumor microenvironment, biomarkers were selected for correlation with KSHV latency-associated nuclear antigen (LANA) expression. The cell surface glycoprotein genes identified were KDR, FLT4, ADAM12, UNC5A, ZP2, and OX40, as well as the endothelial lineage determinants Prox-1 and CD34. Each protein was evaluated for its expression and co-localization with KSHV LANA using multi-color immunofluorescence in KS tissues, KSHV-infected L1T2 cells, uninfected TIVE cells, and murine L1T2 tumor xenografts. Five surface glycoproteins (KDR, FLT4, UNC5A, ADAM12, and CD34) were associated with LANA-positive cells but were also detected in uninfected cells in the KS microenvironment. *In vitro* L1T2 cultures showed evidence of only FLT4, KDR, and UNC5A, whereas mouse L1T2 xenografts recapitulated human KS cell surface expression profiles, with the exception of CD34 and Prox-1. In KS tumors, most LANA-positive cells co-expressed markers of vascular as well as lymphatic endothelial lineages, suggesting KS-associated dedifferentiation to a more mesenchymal/progenitor phenotype.

## 1. Introduction

Kaposi Sarcoma-associated herpesvirus (KSHV), the etiologic agent of Kaposi Sarcoma (KS), pulmonary effusion lymphoma (PEL), multicentric Castleman’s disease (MCD), and KSHV-induced inflammatory cytokine syndrome (KICS), was discovered in 1994 by Chang and Moore [1,2]. The incidence of KS varies depending on geographical location, lifestyle choices, and HIV-1 coinfection status [3,4]. KS occurs at a high prevalence in sub-Saharan Africa (SSA), isolated regions of South America, and the Mediterranean basin [5,6]. KS presents in four distinct forms: classical KS, observed in elderly Mediterranean men; iatrogenic KS, associated with therapeutic immunosuppression in organ transplantation; endemic KS (EnKS), in African individuals without human immunodeficiency virus (HIV-1) co-infection; and epidemic KS (EpKS), in the HIV-1 infected [3,7]. Based on clinical presentation during the early HIV epidemic, where KS became the most prevalent cancer in the HIV-infected individuals, EpKS was considered an acquired immune deficiency syndrome (AIDS)-defining disease [8]. The widespread use of ART treatment, including in SSA, and the resulting increase in HIV-1 suppression has led to decreased EpKS incidence in the US and Europe. Yet, both EpKS and EnKS continue to occur at a high incidence. It is still one of the most prevalent cancers in SSA, and EpKS continues to present at a high incidence in the HIV-infected, despite plasma viral load suppression [6,9].

KS is a highly vascularized cutaneous lesion characterized by the slit vasculature and spindle morphology of growth deregulated cells, which are seemingly of endothelial origin [10,11]. Previously published bulk transcriptomic data from 24 KS tumors compared to cognate normal skin biopsies revealed an increased expression of transcripts related to the vascular endothelial growth factor (VEGF) pathways, namely, FLT4 and KDR, which are receptors for VEGF C/D and B, respectively [12,13]. The upregulation of FLT4 has also been observed in some KSHV-infected endothelial cell lines [14]. *In vitro* findings from KSHV-infected primary endothelial cells linked the expression of FLT4 with an increased expression of the transcription factor Prox-1, a lineage-defining marker of lymphatic endothelial cells [15,16]. An alternate endothelial cell marker CD34, also found on hematopoietic stem cells, is known to be robustly expressed in KS lesions, [17,18]. These heterogeneous findings raise questions about the cellular origin of KS tumor cells and the tissue-specific driver pathways of tumorigenesis.

Despite KSHV being discovered over 25 years ago, models of sustained infection are few, and those of tumorigenesis are even more limited. Fixed human tissues are not conducive for mechanistic studies or for evaluating biomarkers for therapeutic potential; thus, we are exploring the validity of KSHV-infectable telomerase immortalized vein endothelial cells (TIVE) and the long-term KSHV-infected L1T2 cell line [19,20] in recapitulating *in vivo* KS tumor microenvironment characteristics. Consistent with observations from human KS lesions, tumors derived from L1T2 cells have variable percentages of KSHV-infected spindle cells throughout the tumor [20]. Here, we explore the extent to which tumor cell surface marker phenotypes in L1T2 xenografts reflect those from human KS lesions.

We describe the identification of a panel of highly upregulated cell surface glycoproteins from data mining of our previous comparative human KS transcriptomics as well as the validation and expression of the identified panel in KS tissues, KSHV-infected cell lines *in vitro*, and cell line explants in NSG mice. We also explore the alternative endothelial cellular origins of KS lesions and demonstrate the extent to which explant KSHV-infected cell lines and explant models recapitulate the human KS microenvironment and expression patterns.

## 2. Materials and Methods

### 2.1. Patient Sample Collection

Approval to conduct this study was obtained from the review boards of Tanzania National Institute for Medical Research, Ocean Road Cancer Institute, University of Zambia Biomedical Research Ethics Committee; the University of Nebraska-Lincoln (UNL); and the Louisiana State University Health Sciences Center—New Orleans IRBs. Sample collection for this study was as described in Lidenge et al. [12]. Informed consent was obtained from all subjects involved in the study.

### 2.2. Compliance

All animal experiments for generating KS cell line xenograft models were reviewed and approved by the University of Nebraska-Lincoln (UNL) Institutional Animal Care and Use Committee (IACUC) protocol (ID 1812). All personnel who were involved with the mice studies underwent training in appropriate methods to accomplish these procedures.

### 2.3. Cell Culture

L1T2 cells, obtained from ATCC, were cultured in DMEM supplemented with 10% FBS and 1% penicillin/streptomycin and passaged every 3–4 days. At the onset of this project, L1T2 cells were available for purchase from ATCC but are no longer available despite having been demonstrated to maintain long-term KSHV infection and to induce tumors upon xenotransplantation into immune deficient mice [20]. TIVE cells, kindly provided by Rolf Renne, were cultured in Medium 199 supplemented with 20% FBS, 1% penicillin/streptomycin, 1% 200 mM L-Glutamine, and 30 mg Endothelial Cell Growth Factor (Sigma, Burlington, MA, USA) and reportedly lack the capacity to produce tumors in immunodeficient mice [19]. The media were changed two times per week, and the cells were split 1:2 when the culture reached ~60% confluency.

### 2.4. L1T2 Murine Xenograft Generation

The protocols for the preparation and xenografting of mice with human tissues or cell lines have been previously published [21]. Briefly, 6–8-week-old *NOD-scid IL2Rgamma^null^* (NSG) mice were acquired from Jackson Labs, Bar Harbor, ME, USA (Stock Number 005557). The animals were housed in negative pressure individually HEPA-filtered microisolator racked caging. Environmental controls in-cage and in-facility were monitored and maintained by a centralized computer network. All procedures were carried out in a Class II Type A2 biological safety cabinet. To maintain a liquid state, 100 µL of growth factor-depleted Matrigel (BD Biosciences, San Jose, CA, USA) was mixed with 1 × 10^6^ L1T2 cells (ATCC^®^ VR1802, Manassas, VA, USA) in 100 µL cold-PBS and injected subcutaneously into the flank of each irradiated NSG mouse. The mice were observed daily for palpable tumors. Tumor growth was followed by caliper measurements of volume (length, depth, and width) until sacrifice. The mice were anesthetized with isoflurane (5% induction, 2% maintenance) during the xenograft transplantation process, and then tumor growth was monitored by a caliper. During euthanasia, the tumor and major organs such as the spleen, liver, and lung were excised; half of each tissue was frozen in liquid nitrogen immediately after dissection, and the other half was fixed in 10% neutral-buffered formalin for pathological and molecular analyses.

### 2.5. Immunohistochemistry (IHC)

Formalin-fixed, paraffin-embedded (FFPE), KS tumor biopsy specimens were sectioned into adjacent 6 μm-thick sections for protein detection, as described in Tso et al. [22]. Briefly, the slides were deparaffinized in xylene and rehydrated in graded ethanol washes, followed by 30 min treatment with 2% hydrogen peroxide in methanol to quench endogenous peroxidase activity. The slides then underwent antigen retrieval in sodium citrate solution pH 6.0 for 15 min at 98 °C. The slides were blocked for 30 min using Bloxall (Vector Labs, Newark, CA, USA), followed by primary antibody application and overnight incubation at 4 °C. The antibodies used for IHC were as follows: mouse anti-LANA (Leica, Deer Park, IL, USA, 1:100), rat anti-LANA (Abcam, Boston, MA, USA, 1:100), mouse-anti CD34 (Invitrogen, Boston, MA, USA 1:200), rabbit-anti Prox-1 (Abcam, 1:500), rabbit-anti KDR (ProteinTech, Rosemont, IL, USA, 1:1500), rabbit-anti FLT4 (Invitrogen, 1:800), rabbit-anti ADAM12 (Invitrogen, 1:500), rabbit-anti UNC5A (ProteinTech, 1:300), rabbit-anti OX40 (Novus, Centennial, CO, USA 1:500), rabbit-anti ZP2 (Invitrogen, 1:5000). The slides were then washed, and the species-appropriate HRP-labeled polymeric secondary reagent was applied for 30 min at RT, followed by the addition of diaminobenzidine for chromogen deposition. Hematoxylin was used for counterstaining.

### 2.6. Dual-Immunofluorescence (IF)

Dual IF was used to visualize protein colocalization. As described in the IHC protocol, adjacent slides were deparaffinized, followed by antigen retrieval in sodium citrate solution for 15 min at 98 °C, and then blocked for 30 min using Bloxall (Vector Labs). Primary antibodies were added, and the slides were incubated at 4 °C overnight. The primary antibodies used are as follows: rat anti-LANA (Abcam, 1:100), mouse-anti CD34 (Invitrogen, 1:200), rabbit-anti Prox-1 (Abcam, 1:500), rabbit-anti KDR (ProteinTech, 1:200), rabbit-anti FLT4 (Invitrogen, 1:100), rabbit-anti ADAM12 (Invitrogen, 1:100), rabbit-anti UNC5A (ProteinTech, 1:100), rabbit-anti OX40 (Novus, 1:100), and rabbit-anti ZP2 (Invitrogen, 1:100). After washing, the slides were incubated with their corresponding secondary antibody (chicken anti-rat AF647 (Invitrogen, 1:100), donkey anti-mouse AF488 (Invitrogen, 1:100), donkey anti-rabbit AF488 (Invitrogen, 1:100), donkey anti-rabbit AF647 (Invitrogen, 1:100)) for 2 h at RT. The slides were then counterstained with 300 nM DAPI for 30 min at RT and mounted using Fluoro-gel (Electron Microscopy Sciences, Hatfield, PA, USA).

### 2.7. Imaging

IHC imaging was performed using the Motic EasyScan One. IF imaging and cell quantification were performed using the BZX Fluorescence Microscope and the Keyence Hybrid Cell Quantification software version 1.1.1.8.

### 2.8. Statistical Analysis

Pearson’s correlation coefficients were used to identify transcript correlations. Non-parametric Kruskal–Wallis with Dunn’s multiple comparisons tests were used in GraphPad Prism v9 to compare protein expression within KS tumors.

## 3. Results

### 3.1. Identification of Glycoprotein Transcripts That Correlate with KSHV LANA

Using the transcriptomic dataset described in Lidenge et al. (2020) [12], we identified six transcripts that were uniformly upregulated by more than sixfold in all KS tumors as compared to cognate normal skin controls (Figure 1A,B). These transcripts encode empirically demonstrated or informatically implied membrane-associated proteins. Their expression was correlated with the presence of KSHV latently infected cells, as measured by LANA expression. The expression of LANA was correlated with the extent of KSHV infection (KSHV viral burden) within the KS tumor (r = 0.6166). Despite variable transcriptomic expression (on average 1.40-fold upregulated), CD34 was included for lineage discrimination since it has been detected previously in KS lesions at the protein level and is a marker for vascular endothelial cells and hematopoietic stem cells [18,23]. Prox-1, a nuclear marker for lymphatic endothelial cells, was included for lineage discrimination; it was upregulated 21.52-fold in KS tumors and highly correlated with KSHV LANA [16]. Prox-1 has also been implicated in KSHV maintenance in cell culture models and may also contribute to lytic reactivation [15]. KDR, also known as VEGF receptor 2, and FLT4, also known as VEGF receptor 3, were both highly upregulated by 6.19- and 12.60-fold, respectively. CD34, Prox-1, FLT4, and KDR are associated with endothelial cell signaling, vascularization, and angiogenesis during tumor formation [3,11,15,24].

Additional notable upregulated transcripts encoding cell surface proteins were the metalloproteinase ADAM12 (7.19-fold upregulated), the netrin receptor UNC5A (16.89-fold upregulated), the tumor necrosis factor receptor OX40 (6.59-fold upregulated), and the major subunit of the zona pellucida sperm receptor ZP2 (7.71-fold upregulated). UNC5A, OX40, and ZP2 were highly correlated with KSHV LANA expression; however, ADAM12, despite the strong upregulation in KS versus uninvolved skin, showed no significant correlation with LANA and the KSHV latency program. This short list of transcriptomically implied cell surface markers of KS was evaluated for protein expression in human KS lesions, TIVE and L1T2 cell lines, and L1T2-derived mouse xenograft tumors to determine whether the proteins were KSHV-infection-associated or markers of the KS tumor microenvironment.

### 3.2. Validation of Protein Expression in KS Tissues

It has been well demonstrated that transcript expression values calculated from bulk transcriptomics do not always accurately reflect protein abundance in tissues [25,26]. Thus, we sought to determine the extent of the expression and localization of each protein in KS tumors using immunohistochemistry. KS tumor biopsies were sectioned and stained for LANA expression to identify regions with demonstrable KSHV infection (Figure 2A). Representative images from adjacent slides are shown in Figure 2B–K for each identified protein and isotype control. KDR, FLT4, Prox-1, CD34, ADAM12, and UNC5A were all robustly expressed throughout the KS tumors. Staining intensity was similar across multiple lesions regardless of the extent of LANA expression or its intensity, both of which were variable across KS lesions [27]. OX40 and ZP2 were not detected, which is consistent with the lack of T cells in KS lesions and with the highly restricted, tissue-specific expression and function of ZP2, respectively [27]. The efficacy of the ZP2 and OX40 antibodies was confirmed using human ovary and human lymph node tissues, respectively. Protein expression was also evaluated in normal skin using dual immunofluorescence, where low to no staining of the identified markers was detected (Appendix A).

### 3.3. Highly Expressed Proteins Are Not Exclusive to KSHV-Infected Tumor Cells

Having demonstrated the upregulated expression of several transcriptomically implicated cell surface proteins in regions of KS tissue harboring demonstrably KSHV-infected cells, we next sought to determine whether those proteins were directly co-localized with LANA (i.e., KSHV infection) or, alternately, if they were upregulated cell markers in the tumor environment. We conducted dual immunofluorescence assays with each identified protein and LANA. The same antibody clones used for IHC were used for IF to minimize sensitivity differences based on epitope recognition between approaches. Representative images are shown in Figure 3A–F. The surface-associated proteins, KDR, FLT4, CD34, UNC5A, and ADAM12, were all robustly expressed throughout the lesions but were not exclusive to LANA-positive cells. Prox-1, a protein localized in the nucleus, was detected in most LANA+ cells but was also observed in some LANA– cells (Figure 3C). Consistent with the IHC results, OX40 and ZP2 remained undetectable by dual IF (Appendix A).

### 3.4. Cells in KS Lesions Co-Express Prox-1 and CD34

We next analyzed the frequency of the co-localization of Prox-1 and CD34, proteins which are normally part of distinct endothelial cell lineages and, therefore, are not typically expressed on the same cells. We evaluated 16–20 fields from 5 different lesions (N = 83) for staining patterns of Prox-1, CD34, and LANA expression. This revealed that 25% of tumor cells were CD34+/Prox-1+, 26% CD34+ only, and 4% Prox-1+ only (Figure 4A,E). We also calculated the degree to which the endothelial markers were colocalized with KSHV infection, as denoted by LANA staining. The KS tumors were 22% Prox-1+/LANA+, 14% Prox-1+ only, and 7% LANA+ only. We also observed that 66% of the cells in the KS tumors were CD34+, but only 34% of the tumor cells were also LANA+, and less than 6% were LANA+ only (Figure 4B,C,F,G). The five KS tumors all varied in the percentage of LANA-positive cells, which highlights the heterogeneity of KS lesions when evaluating for biomarkers (Figure 4D). The cell counting software does not allow for the quantification of three markers at once; however, staining for LANA, CD34, and Prox-1, as shown in Figure 5A–F, demonstrated a mixture of cell phenotypes and variable staining patterns. Together, IHC staining and IF colocalization data suggest that the majority of cells in KS tumors co-express markers of both vascular endothelial and lymphatic endothelial lineages, suggesting that the lesions are lineage indistinguishable or may have dedifferentiated to a progenitor-like phenotype.

### 3.5. Limited Protein Expression in KSHV-Infected Endothelial Cell Lines

Since mechanistic studies for investigating potential roles for these cell surface proteins in defining and contributing to the KS tumor microenvironment require model systems that faithfully mimic the *in vivo* (human) tumor microenvironment, we then tested the KSHV-infected cell line, L1T2, and its uninfected endothelial cell counterpart, TIVE. L1T2 is a heterogeneous cell line derived from KSHV-infected TIVE cells implanted into mice, as described in Roy et al. (2013), where it was reported that tumors reminiscent of KS develop and recapitulate the protein expression phenotypes of KS tumors [20]. There was a low but detectable protein expression of KDR, FLT4, and UNC5A in the *in vitro* cultured L1T2 cells but not in the uninfected TIVE cells, which were uniformly negative (Figure 6A–F). The endothelial lineage markers CD34 and Prox-1, as well as ADAM12, were not detectable in either L1T2 or TIVE cells (Figure 6G–L). We also observed an overall low frequency of KSHV-infected cells in the L1T2 culture. Isotype controls are seen in Appendix A. Since only three out of six markers readily detected in human KS tumors were detected in L1T2 *in vitro*, this finding, coupled with the lack of transcriptomic concordance between KS and cell infection models, highlights the challenges and limitations of recapitulating KS tumor biology with cell culture models of KSHV infection and KS tumorigenesis [13].

### 3.6. L1T2-Derived Xenografts Reflect the Protein Expression Observed in Human KS Tissues

Nevertheless, L1T2 cells readily form KS-like lesions with increased numbers of KSHV-infected cells when transplanted into mice; thus, we tested for the presence of the KS tumor microenvironment cell surface markers to evaluate the extent to which xenografts recapitulate the human KS tumor environment. The FLT4, KDR, and UNC5A proteins were detected at high levels within the L1T2 xenografts, mirroring the human KS tumor expression patterns (Figure 7A–C). Surprisingly, despite not being detected in the L1T2 cell line, ADAM12 was also highly expressed in xenografted L1T2 (Figure 7D). As with L1T2 *in vitro* cultures, CD34 and Prox-1 were not detected in the xenografts, suggesting that the L1T2 cell line and mouse xenografts, despite KSHV infection, are not differentiated endothelial cells (Figure 7E,F). We also evaluated the expression of LANA and the human surface markers in the spleens of the tumor-bearing mice, where no cross-reactive staining was observed except for low levels of ADAM12 (Appendix A). Isotype controls for the L1T2 xenografts are shown in Appendix A. In summary, we detected the partial recapitulation of KS tumor microenvironment cell surface protein expression within the xenograft tissues where FLT4, KDR, UNC5A, and ADAM12 mirrored the levels detected in the human KS tumors. However, the lack of endothelial lineage markers in both pre-implant and post-implant L1T2 cells may limit the applicability of this model for the investigation of KS origins.

## 4. Discussion

The lack of *in vivo* models that accurately reflect the phenotypes and behaviors of cells in human KS lesions is an impediment to studies of KSHV infection, KS tumorigenesis, and pathogenesis, as well as to the preclinical evaluation of potential treatments. To address these impediments, and to further refine our understanding of the KS tumor microenvironment, we utilized previously published gene expression profiles from 24 KS lesions, each subtractively compared with autologous non-involved skin. We identified a subset of seven highly upregulated transcripts that correlated with KSHV LANA expression, six of which were predicted to be present on the cell surface of at least some component lineage within KS tumors. These cell surface markers might be potential biomarkers of latent infection with KSHV or of KS-transformed cells and could be potential targets for KS therapy. Alternately, these markers, which were not elevated in uninvolved skin, might be induced in bystander cells by the processes of infection or transformation. Other markers, such as the transcription factor Prox-1, were investigated for their potential to refine our appreciation of the specific endothelial cell lineage(s) that comprise KS tumors. Here, in correlation with transcriptome data, we demonstrated the detectable protein expression of KDR, FLT4, UNC5A, ADAM12, CD34, and Prox-1 in the human KS tumor microenvironment. Despite significantly upregulated RNA expression in KS tumors, neither OX40 nor ZP2 proteins were detected. The absence of the former is consistent with its typical expression as a co-stimulatory molecule on T cells, which we have reported as largely absent in the vicinity of KSHV-infected cells in KS biopsies [27]. The failure to verify ZP2 is likewise unsurprising given that it is typically expressed almost exclusively in the human ova [28]. Thus, both upregulated transcripts may undergo some form of post-transcriptional regulation that limits/prevents their translation in the constituent cells of the KS microenvironment, as has been demonstrated in transcriptomic–proteomic decoupling in other studies [25,29].

ADAM12, Prox-1 and CD34, were expressed at elevated levels in KS lesions versus normal skin. We compared the extent to which the KSHV-uninfected TIVE, KSHV-infected L1T2 cells, and the L1T2-derived mouse xenograft tissues recapitulated the KS tumor protein expression phenotypes. In contrast to KS lesions, ADAM12, Prox-1 and CD34 were not detectable in either the TIVE or L1T2 cell lines; however, ADAM12 was evident via IF in L1T2 xenografts. The metalloproteinase ADAM12 has been shown to be connected to hypoxia as well as T-cell mediated TGFβ activation, leading to cell death and sustained inflammation [30,31]. The activation of TGFβ has been demonstrated to be through the stabilization and recruitment of TGFβ receptor II, which was also upregulated in our KS transcriptomics [32]. The lack of T-cell infiltration in KS lesions and the robust ADAM12 expression in both human KS lesions and the L1T2 mouse xenografts make ADAM12 an interesting potential target and biomarker.

Despite some lineage marker data supporting both vascular and lymphatic endothelial origins of KS, this remains speculation due to inconsistency in systems and the lack of a *de novo* transformation model. Evidence from *in vitro* infections of vascular and lymphatic endothelial cells suggests that lymphatic endothelial cells (Prox-1+/CD34−) are more susceptible to KSHV infection and long-term episomal maintenance than vascular endothelial cells (Prox-1−/CD34+) [15]. However, in the current study, we found that 25% of the KS tumor cells were co-expressing CD34 and Prox-1 which implies two scenarios that cannot, at this time, be distinguished. In the first, dedifferentiation events occurring during infection or tumorigenesis could result in an infected cell type with a more primordial/progenitor endothelial lineage lacking distinct lineage commitment markers. This type of dedifferentiation has been reported in other cancers and cancer virus infections [33,34]. An alternate concept involves the KSHV infection of mesenchymal stem cells that undergo differentiation or perhaps variable differentiation to endothelial phenotype(s) based on other factors [10,35]. This transition has been suggested to be associated with the KSHV-driven activation of Prox-1 [15]. The co-expression of CD34 and Prox-1 may support this concept but might suggest an incomplete or heterogeneous switch from mesenchymal to endothelial cells, where heterogeneous protein expression phenotypes may derive from unique and, to date, undefined microenvironmental determinants in KS tumors. Colon cancer has also been shown to contain endothelial cells which co-express both CD34 and Prox-1, which may suggest that the co-expression of these lineage markers is a result of tumorigenesis in endothelial tumors [36]. However, currently, it remains unclear in KS whether the microenvironment defines the tumor phenotype or, rather, the tumor phenotype defines the tumor microenvironment.

There is much interest in generating *in vivo* models that more completely reflect the human KS tumor environment since endothelial cell lines, the most time- and cost-effective choice, have been shown to only reflect about 10% of the transcriptomic changes seen in KS lesions [13]. Here, we revealed that while TIVE cells lack the cell markers of KS, the KSHV-infected L1T2 cell line derived from them did not fully gain protein expression phenotypes evident in KS, since only 50% of the validated protein markers were evident. *In vivo* animal models may be more likely to recapitulate the KS microenvironment; however, tumor generation has proven difficult and inconsistent using both cell line and human KS tumor xenografts. Furthermore, while FLT4, KDR, UNC5A, and ADAM12 were detected in the L1T2 mouse xenografts in concert with human tumor results, we did not detect either endothelial cell marker (Prox-1 and CD34) in those grafts, which may indicate that the L1T2 cell line has lost its endothelial cell attributes in culture. Thus, as we have shown in the comparison of human transcriptomics to those from infected cell lines, protein expression profiles are also differential between KS patient tumors and cell lines/mouse models.

The consistency in the transcriptomic expression of the selected markers indicated that our sample size is robust enough to be representative and generalizable. Consistency was also evident at the protein level, where the detectable markers were consistently found in all tissues evaluated, albeit with variation in the signal magnitude from field to field depending on the cellularity, the cell types, and the number of KSHV-infected cells. Additionally, many of these markers, namely, KDR, FLT4, and UNC5A, have been shown to be involved in other types of cancers, making these markers non-exclusive to KS tumorigenesis [11,24,37]. The specific role of these proteins in KS lesions has yet to be fully explored. The two VEGF receptors, KDR and FLT4, are currently being explored for targeted inhibition therapy to reduce immune modulation and immunosuppression in the tumor environment [38]. These studies have, to date, produced inconclusive results and require more exploration with *in vivo* models to ascertain the effects of therapy on KS tumorigenesis.

## 5. Conclusions

We show the robust expression of potential candidate therapeutic targets, KDR and FLT4, as well as newly identified biomarkers, UNC5A and ADAM12, expressed consistently in KS tissues. Our current study also highlights the importance of validating transcriptomic data at the protein level, since the OX40 and ZP2 glycoproteins were not detectable despite 6.59- and 7.71-fold upregulation in gene expression, respectively. We also showed, through the robust upregulation of the expression of KDR and FLT4 in L1T2 versus TIVE cells, that L1T2-derived mouse xenografts could be a model for studying pathways, such as VEGF activation in KSHV infection and neoplasia, but they cannot be used to resolve the endothelial origin of KS, as they lack appropriate lineage markers. Further evaluation in additional cell lines, lineages, and animal models is needed to identify the conditions that more completely recapitulate the human KS tumor microenvironment and the landscape of the cellular membrane proteins.

## Figures and Tables

**Figure 1 cancers-15-02171-f001:**
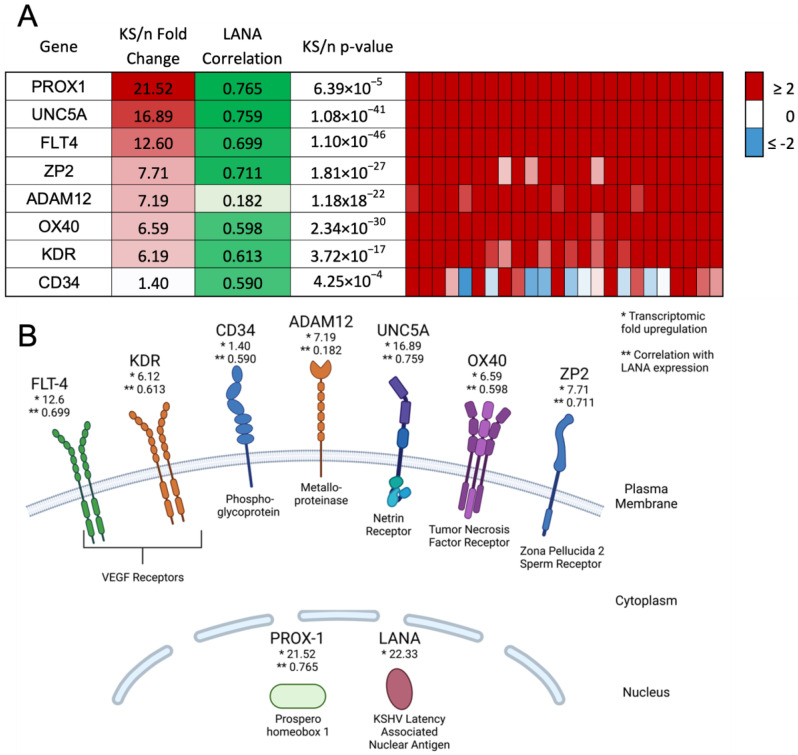
Identification of potential biomarkers with high upregulation in KS tumors and high correlation with LANA expression. (**A**) Overall fold change, correlation with LANA expression, and *p*-value are shown, as well as transcript expression within each individual KS lesion. (**B**) Diagram indicating the cellular location of markers at the protein level. Created using BioRender.com. * indicates the transcriptomic fold upregulation of KS tissues over normal human skin. ** indicates the correlation with LANA expression in the KS lesions.

**Figure 2 cancers-15-02171-f002:**
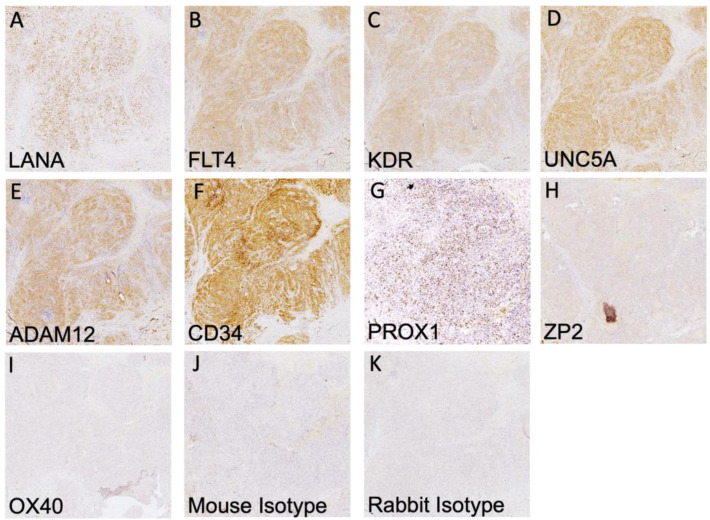
IHC detection of all human cell markers and KSHV LANA. (**A**–**G**) Positive staining is indicated by HRP staining within human KS tissues of the indicated surface marker and KSHV LANA protein. (**H**,**I**) There was no detection of ZP2 and OX40 proteins despite robust transcript detection. (**J**,**K**) Mouse and rabbit isotype controls. IHC images were taken at 10X magnification.

**Figure 3 cancers-15-02171-f003:**
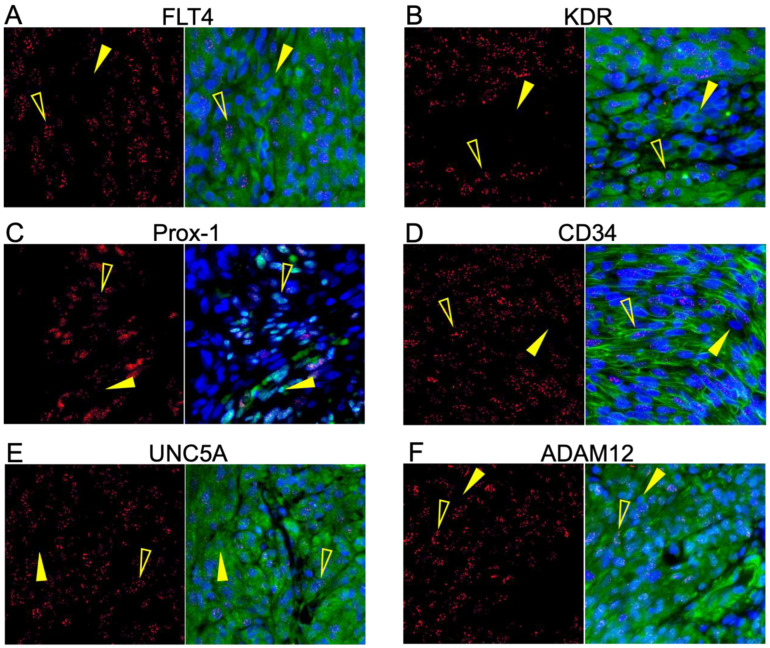
Six of the potential markers are robustly translated into protein within KS tumors. (**A**–**F**) Dual IF of the co-expression of LANA and the indicated potential KS biomarker. FLT4, KDR, UNC5A, ADAM12, CD34, and Prox-1 had robust expression in the KS lesions and were not exclusively detected in LANA-positive cells. ZP2 and OX40 were not detected at the protein level in KS lesions despite high transcriptomic expression. LANA is denoted as red, and the surface marker in question is green. Open yellow arrows indicate LANA-positive cells, and closed arrows indicate LANA-negative cells.

**Figure 4 cancers-15-02171-f004:**
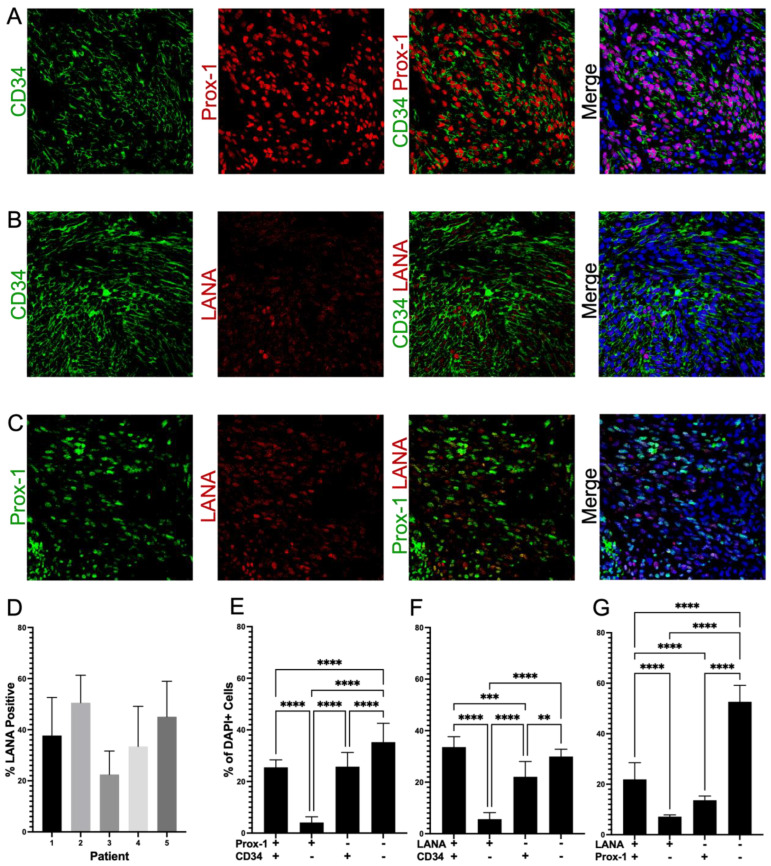
Colocalization of distinct endothelial lineage markers suggests KS tumors are dedifferentiating to an HSC phenotype. (**A**) Colocalization of Prox-1 and CD34, (**B**) Colocalization of LANA and CD34, and (**C**) Colocalization of Prox-1 and LANA. (**D**) Variation in the percentages of LANA-positive cells in five different KS lesions, which contributes to the quantification variation of cellular phenotypes across tissues. (**E**–**G**) Quantification of expression within KS tumors; about 25% of the tumor is double-positive for Prox-1 and CD34; greater than 50% of the tumor is CD34-positive, but only about 50% of these cells are also LANA-positive. Images were acquired at 20X magnification. Non-parametric Mann–Whitney analysis was used to determine statistical differences. Significant levels are *p* < 0.01 **, *p* < 0.001 ***, *p* < 0.0001 ****.

**Figure 5 cancers-15-02171-f005:**
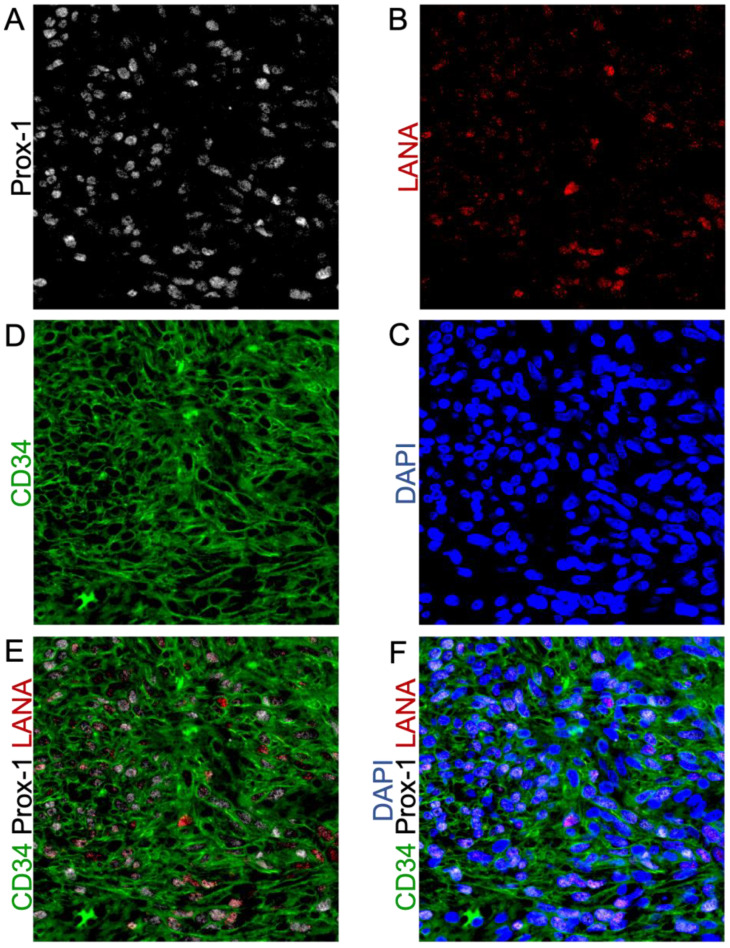
Triple-Stained IF shows the heterogeneous expression of KSHV-LANA and the endothelial lineage markers Prox-1 and CD34. (**A**) Prox-1 staining; (**B**) LANA staining; (**C**) nuclear DAPI staining; (**D**) CD34 staining; (**E**) Co-localization of KSHV-LANA, Prox-1, and CD34; (**F**) Co-localization of KSHV-LANA, Prox-1, and CD34 with nuclear DAPI staining. Images were acquired at 20X magnification.

**Figure 6 cancers-15-02171-f006:**
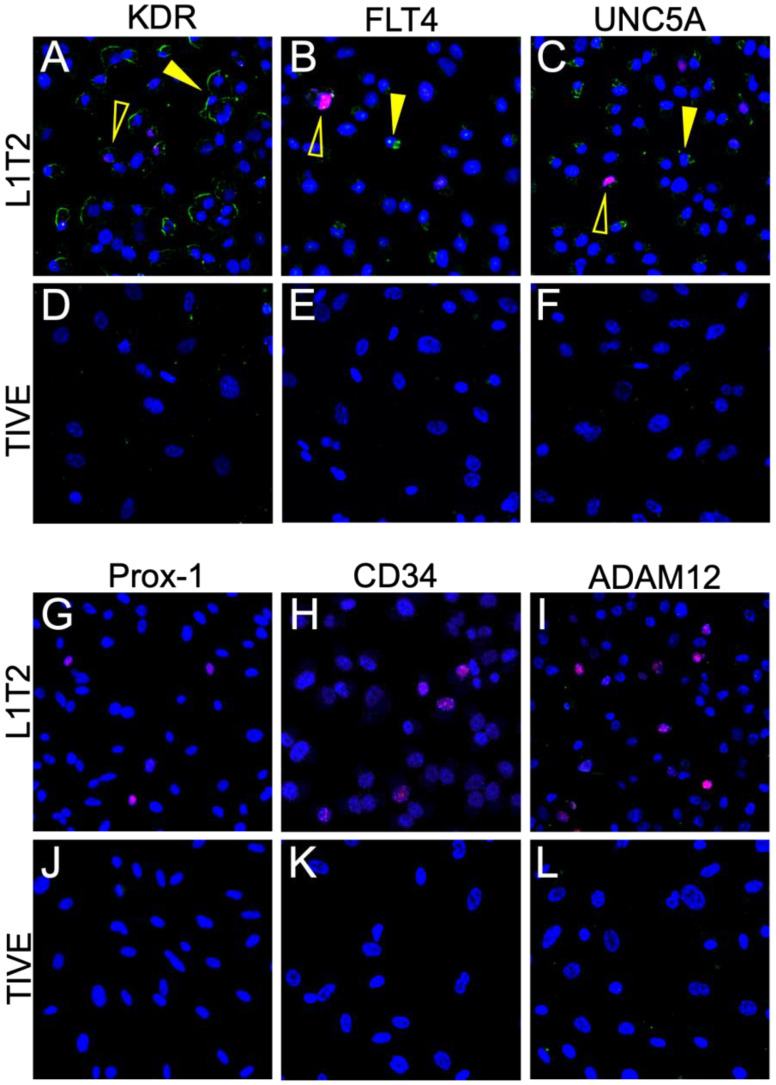
Dual IF of KSHV-infected L1T2 and uninfected TIVE cells shows the expression of KDR, FLT4, and UNC5a but not ADAM12 or the endothelial markers. (**A**–**L**) Dual IF of the co-expression of LANA and the indicated potential KS biomarker. ADAM12, CD34, and Prox-1 were not detected in either cell line. FLT4, KDR, and UNC5A had low to moderate expression in the KS lesions and were not exclusively detected in LANA-positive cells. LANA is denoted as red, and the surface marker in question is green. Open yellow arrows indicate LANA-positive cells, and closed arrows indicate LANA-negative cells. Images were acquired at 20X magnification.

**Figure 7 cancers-15-02171-f007:**
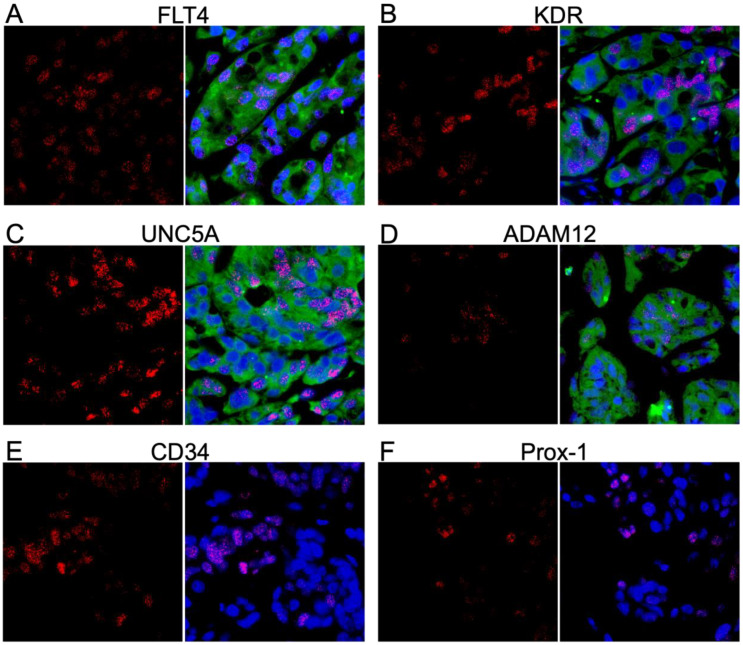
Dual IF demonstrating the colocalization of markers and LANA in mouse xenograft tissue. (**A**–**F**) Dual IF of the co-expression of LANA and the indicated potential KS biomarker. FLT4, KDR, UNC5A, and ADAM12 had robust expression in the KS lesions and were not exclusively detected in LANA-positive cells. CD34 and Prox-1 were not detected at the protein level. LANA is denoted as red, and the surface marker in question is green. Images were acquired at 20X magnification.

## Data Availability

All relevant data are included in the manuscript or Appendix A. Any additional requests can be directed to the corresponding author.

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
