# Peer review of "Upregulation of Cell Surface Glycoproteins in Correlation with KSHV LANA in the Kaposi Sarcoma Tumor Microenvironment"

_cancers, 2023, doi:10.3390/cancers15072171_

Round 1

Reviewer 1 Report

This concise manuscript by Privatt et al is a largely descriptive study of surface markers in KS tumors that were previously identified in a transcriptomic screen, KS model cell lines and xenografts that seeks (1) to describe the presence and distribution of these surface markers and (2) uncover how tumor markers are similar/different comparing KS tumors to the L1T2 in vitro model and L1T2 xenografts. Although there are numerous references to the potential utility of these surface markers as therapeutic targets or biomarkers, there is no data exploiting them as either one. However, this study lays the foundation for such mechanistic studies to be performed. I believe that the data comparing the KS tumor IHC to in vitro and xenograft tumors is important and highlights an area that needs further study in the field. Specifically, that despite a lack of good animal models for KS, purely in vitro models lack critical microenvironmental factors that are key in understanding tumorigenesis. Without mechanistic exploitation of these targets/biomarkers the significance of this study is not very high, but it is generally well-executed and the results can be widely important in the field from a foundational target selection as well as a model selection standpoint. 

Specific comments: 
Major -

Figure 1: because your conclusions are based on these surface markers being specifically upregulated in KS tumors, I believe the relegation of the normal skin controls to "data not shown" is inappropriate. the isotype controls tell us that your staining is specific to the protein, but not that overexpression of these markers is specific to the tumor. At minimum, (preferably patient-matched) normal skin controls should be included as supplemental information.

The observation that Prox1 and CD34 are co-expressed in tumor-associated endothelial cells is not novel or necessarily specific to KSHV-associated tumors (see PMC1606520, for example). The authors should include this in their discussion of the novelty of these findings and also the potential of these changes being viral vs. tumor microenvironmentally driven. 

Reviewer 2 Report

In this manuscript by Privatt et al, the authors use previously published KS tumor transcriptomics to identify cell surface biomarkers to identify potential therapeutic targets. They identified several biomarkers that correlate with LANA expression and validated in KS tissues, infected cells, and mouse L1T2 xenografts. Some biomarkers were detected not only in LANA positive cells but also within the tumor microenvironment. There aren’t many models of sustained KSHV infection, making it hard to study the KS tumor microenvironment, and such models are needed for mechanistic studies evaluating the therapeutic potential of biomarkers. In this study, the authors aim to compare L1T2 derived xenografts with fixed human KS lesions to determine whether this is a good model system for further study of these biomarkers.

Comments:

Methods – L1T2 cells are not available on ATCC as of this writing

For IHC staining - In text – line 213 – authors state no staining in normal skin controls, isn’t that odd that its zero staining, not just reduced staining?

For the IF staining, I see high expression either correlating with LANA or not, which I understand is the point, but I am wondering what expression looks like in the absence of virus. It is hard to tell whether the staining is due to presence of virus or just high anyways. I would like to see uninfected samples and zoom in on the -/+ LANA regions in the current images, so it is more clear. For example, FLT4 is green everywhere and its hard to tell that this staining is specific for KSHV induced FLT4 expression,

Fig 4 colocalization experiments – this is done in 16 fields from 2 lesions. Are 2 lesions sufficient? How many biological replicates are needed to claim that KS tumors dedifferentiating to a HSC phenotype? I would expect at least 3 biological replicates would be needed for statistical significance.

Fig 7 – I would like to see staining in matched mouse tissues for comparison. I see very strong staining in the xenograft, is it zero in normal matched mouse tissue or just reduced.

Overall data is very interesting, but throughout the paper I wanted to see staining of uninfected cells/ matched mouse tissues. Isotype controls are included but they don’t tell me how target protein expression changes from uninfected to infected/tumor/xenograft.

Reviewer 3 Report

Line 172-173: Does the correlation with LANA expression measure latency or overall KSHV viral burden? Would a ratio of latent versus lytic genes be a better marker of latency? Or should part of the title be changed to "Correlation with KSHV LANA"?

Fig. 1 A and Fig. 1B are a bit redundant. 

Fig. 2: What positive controls were used for OX40 and ZP2 staining?

Line 212: It is unclear if the authors mean that none of the proteins in Fig. 2 were detected in normal skin. 

Fig. 6A-C: The staining for LANA appears to be low or absent from the majority of cells. The authors should comment on this observation. Perhaps show separate images with only LANA staining. Flow cytometry might be a better way to look at co-expression with cultured cells.

Fig. 7: The LANA staining appeared to be much more robust in the L1T2 mouse xenograft tissue as compared to the L1T2 cells in Fig. 6. Please explain. 

Reviewer 4 Report

Kaposi’s sarcoma-associated herpesvirus (KSHV) is an oncogenic virus, causative agent of specific human diseases, including cancers, such as Kaposi Sarcoma (KS). This manuscript by Privatt et al. is a straightforward well-constructed study focusing on identification and characterization of novel potential cell surface biomarkers and their association with KSHV in KS tumors. The authors begin their study by systematically dissecting previously published transcriptome data comparing KS samples to cognate normal skin control biopsies, which resulted in the identification of a set of upregulated transcripts in KS correlating with LANA levels. Subsequently, they focused on several such upregulated transcripts encoding membrane-associated factors, related to the VEGF pathway. The authors then investigated the levels of the encoded specific biologically relevant proteins, including glycoproteins (namely FLT4, KDR, UNC5A and ADAM12) in human Kaposi Sarcoma (KS) tissue samples as well as KSHV infected cells and derived mouse xenografts. The study is a well-executed comparative analysis, systematically evaluating the levels of these novel biomarkers in various models. Importantly, the authors were able to demonstrate consistently elevated high protein levels for several newly identified biomarkers in all the sampled KS tissues and also dissected their levels in the various cell lines and grafts.  Importantly, the role of these factors has been recognized previously in other cancer types, but their role remains to be investigated in KS context, thus evaluation of their protein levels in KS is of significance.

Minor points:

1.     Figure legend is missing for the top right panel (for the data showing transcript expression levels within the individual KS lesions) in Fig1A.

2.     The authors refer to “KSHV latency” in the title and throughout the manuscript, yet the authors only address LANA positivity, but do not assay for lytic protein levels. Naturally, a small subset of such LANA positive cells can also be lytic, thus it would be more appropriate to refer to KSHV infection or KSHV+ cells, instead of KSHV latently infected cells for such population in the manuscript. It would be beneficial to add a short sentence to clarify this point for the reader, and changing the statements where appropriate.

Round 2

Reviewer 1 Report

The authors have adequately addressed my comments.

Reviewer 2 Report

Comments were addressed